# Genetic Differentiation of *Aedes aegypti* (Diptera: Culicidae) in Areas with High Rates of Infestation in Mid-North Region of Brazil

**DOI:** 10.3390/insects14060530

**Published:** 2023-06-06

**Authors:** Luzianny Farias Rodrigues, Andrelina Alves de Sousa, Walter Pinheiro Mendes Júnior, Amanda Caroline Cardoso e Silva, Maria Histelle Sousa do Nascimento, Maria Claudene Barros, Iracilda Sampaio, Elmary da Costa Fraga

**Affiliations:** 1Graduate Program in Biodiversity, Environment and Health, Laboratory of Genetics and Molecular Biology, Universidade Estadual do Maranhão—UEMA, Caxias 65604-380, MA, Brazil; luziannyrodrigues@hotmail.com (L.F.R.);; 2Graduate Program in Genetics and Molecular Biology, Laboratory of Genetics and Molecular Biology, Universidade Federal do Pará—UFPA, Belém 66075-110, PA, Brazil; andrelbio@yahoo.com.br (A.A.d.S.);; 3Graduate Program Network Biodiversity and Biotechnology of Legal Amazon, Biological Scienses Institute, Belém 66075-110, PA, Brazil

**Keywords:** dengue fever vector, genetic diversity, microsatellites

## Abstract

**Simple Summary:**

The mosquito *Aedes* (*Stegomyia*) *aegypti* represents a major challenge for public health due to its proliferation in many regions of the world and its capacity to disperse arboviruses. The verification of the genetic characteristics of this mosquito and the differentiation of these characteristics both within and between its populations in the Mid-North region of Brazil will be fundamentally important for the development of effective measures for the control of this disease vector. Previous studies of mitochondrial markers in the Mid-North region revealed two genetic lineages of *Ae. aegypti*, and high levels of genetic differentiation, which represent a potential risk for the occurrence of epidemics and a major problem for local public health authorities. Using microsatellite markers, the present study identified distinct genetic lineages of *Ae. aegypti* in areas with high rates of infestation. This analysis revealed the coexistence of two different genetic lineages, which were highly differentiated in genetic terms. These findings on the genetic status of the populations will be extremely important for the development of innovative strategies for the control of the populations of this disease vector, and effective measures for the protection of local human populations.

**Abstract:**

*Aedes aegypti* is the principal vector of the arboviruses—yellow fever, dengue virus, chikungunya, and zika virus. Given the epidemiological importance of this mosquito, its capacity to adapt to different habitats, and its resistance to many types of control measures, systematic research into the genetic variability of the populations of this mosquito is one of the most important steps toward a better understanding of its population structure and vector competence. In this context, the present study verified the presence of distinct genetic lineages of *Ae. aegypti* in areas with high infestation rates, based on the analysis of microsatellite markers. The samples were collected in nine municipalities with high building infestation rates in the Mid-North region of Brazil. Six microsatellite loci were genotyped in the 138 samples, producing a total of 32 alleles, varying from one to nine alleles per locus in each of the different populations. The AMOVA revealed greater within-population genetic differentiation with high fixation rates. The general analysis of population structure, based on a Bayesian approach, revealed K = 2, with two *Ae. aegypti* lineages that were highly differentiated genetically. These data on the connectivity of the populations and the genetic isolation of the lineages provide important insights for the development of innovative strategies for the control of the populations of this important disease vector.

## 1. Introduction

*Aedes (Stegomyia) aegypti* (Linnaeus, 1762) is an invasive mosquito of the family Culicidae, which is native to Africa but is now widely distributed in the tropical and subtropical regions of the world [1]. This mosquito is a major concern for public health, given that it is the principal vector of the yellow fever arbovirus, all four dengue fever serotypes (DENV-1, DENV-2, DENV-3, and DENV-4), and the zika and chikungunya viruses [2,3,4,5].

The control of this vector is considered to be a major challenge for public health authorities around the world, given the capacity of the mosquito to adapt to different types of habitat, the hardiness of its eggs, passive dispersal, and its resistance to control measures [6,7,8]. These factors have contributed to the high rates of occurrence and the dispersal of arboviruses around the world [2,5].

In Brazil, *Ae. aegypti* was eradicated in the 1950s, according to the Pan American Health Organization (PAHO) but was subsequently reintroduced in the 1970s [9]. Since this time, *Ae. aegypti* has been considered to be one of the vectors that have had the greatest impact on public health in the country. Up until June 2022, for example, 1.172.882 probable cases of dengue fever were recorded in Brazil, in addition to 122.075 cases of chikungunya, and 5.699 of zika [10]. This scenario is further aggravated by the discovery of the occurrence of a new lineage of the zika virus in the country, which may mean a more virulent reemergence of this arbovirus in the near future [11].

These findings reinforce the need for the genetic monitoring of *Ae. aegypti* populations. The epidemiological importance of the species demands reliable data on its population structure. The SSRs are good molecular markers for the study of population genetics, given their high rates of mutation and polymorphism [12,13]. In particular, SSRs are low-cost, neutral markers that, in recent years, have provided valuable insights into genetic diversity, gene flow patterns, and intra- and inter-specific genetic differentiation in a range of studies, both in Brazil and around the world [14,15,16,17].

Studies of the genetic variability of *Ae. aegypti* populations are extremely important for the Mid-North region of Brazil, where a large proportion of the municipalities are at risk of infestation by this vector and are thus faced with the potential spread of the dengue, chikungunya, and zika arboviruses [10,18,19,20,21,22]. In two previous studies in Maranhão, one of the Brazilian states that make up the Mid-North region, Fraga et al. [23] and Sousa et al. [24] found high levels of intra-population genetic differentiation in *Ae. aegypti*, with evidence of the coexistence of two distinct genetic lineages in the state. It is important to note that populations of vectors with high levels of genetic differentiation within a given (micro or macro) region may present a greater competence for the transmission of diseases.

Given this, it will be essential to compile reliable data on the patterns of genetic structure and gene flow and the within- and between-population differentiation of *Ae. aegypti* in order to support the development of effective measures for the control of these populations within a given area. In this context, the present study builds on the results of three previous population studies of *Ae. aegypti*, two of which [23,24] focused on mitochondrial markers in the state of Maranhão, while the third [16] analyzed microsatellite markers in the Amazon Region. Here, microsatellite markers were genotyped in order to determine the possible presence of distinct genetic lineages in municipalities with high *Ae. aegypti* infestation rates in the Mid-North Region of Brazil.

## 2. Materials and Methods

### 2.1. Study Area and Sample Collection

The Mid-North (Meio-Norte) is a region in the western extreme of Northeastern Brazil, which encompasses the ecological transition between the Amazon and Cerrado biomes, including the state of Maranhão and the western half of Piauí [25]. The samples were collected in nine municipalities of the Mid-North region—six in Maranhão and three in Piauí (Figure 1)—that are known to have high Building Infestation Rates (BIRs), based on the results of the Rapid Index *Ae. aegypti* Survey (LIRAa) provided by the Brazilian Ministry of Health [18,19,26]. The details of the study areas and the collection of samples are shown in Table 1.

We obtained the samples using 20 ovitraps [27], which were distributed in different neighborhoods of each study municipality. To avoid collecting related individuals, the traps were set at distances of between 20 m and 5 km from each other [9]. Each trap was set in a shaded, poorly lit area adjacent to residential buildings and was retrieved after six days. The pallets containing eggs were taken to the Laboratory of Genetics and Molecular Biology at the Caxias campus of Maranhão State University (UEMA), where they were placed in individuals containers with water to allow for the hatching of eggs and the development of larvae to the L4 stage, following the protocol of Santos et al. [28].

All the biological specimens were identified as species using the taxonomic key of Consoli and Lourenço-de-Oliveira [1]. Once identified, the specimens were placed in 1.5 mL Eppendorf microtubes containing 70% ethanol, then frozen in a freezer at −80 °C. To avoid sampling siblings, only one larva from each container, and consequently, from different pallets, was selected for the molecular procedures.

### 2.2. Extraction and Amplification of the DNA, and Genotyping 

The total DNA was extracted from the specimens using the Wizard Genomic DNA Purification kit (Promega, Madison, WI, USA), following the manufacturer’s instructions. The six polymorphic loci were amplified by Polymerase Chain Reaction (PCR) using the specific primers (Table 2) described by Huber et al. [29] and Chambers et al. [30]. The loci were amplified in a final volume of 7 μL containing 0.7 μL of the DNA (50 ng/μL), 3.5 of Master Mix-PCR Multiplex (QIAGEN, Germantown, MD, USA), 0.35 μL of the forward primer (100 μM/μL), 0.7 μL of the reverse primer (100 μM/μL), 0.35 μL of fluorescent dye (100 μM/μL), and 1.4 μL of ultrapure water. The amplification protocol of the microsatellite loci consisted of initial denaturation for nine minutes at 9 °C, followed by a second stage of 35 cycles of 94 °C for 30 s (denaturation), 57.5 °C for 90 s (annealing), and 72 °C for 90 s (extension), then a third stage of 15 cycles of 94 °C for 30 s (denaturation), 53 °C for 45 s (annealing), and 72 °C for 45 s, with a final extension of 5 min at 72 °C.

The PCR products were diluted by 10× in ultrapure water, with 0.3 μL of 600 LIZ Size Standard and 8.7 μL of formamide being added to 1 μL of this product. The plates containing 10 μL of each sample were placed in a thermocycler and heated to 95 °C for 5 min, after which the samples were sequenced in an ABI PRISM 3500 (Life Technologies, Carlsbad, CA, USA) automatic DNA sequencer, following the manufacturer’s recommendations.

### 2.3. Data Analysis

The results of the genotyping were analyzed in the GENEMARKER program [31] to verify potential patterns in the distribution of the alleles and the genotypes, which were inserted into an Excel spreadsheet to establish the database. The input files for the subsequent analyses were converted and created in CREATE 1.33 [32]. The MICRO-CHECKER 2.2.3 program [33] was used to analyze the presence of null alleles in the loci. Genetic variability was verified in GENALEX 6.5 [34], which was used to determine the frequency and distribution of the alleles per locus and per population, and the effective number of alleles Ne(allele), and the observed (Ho) and expected heterozygosity (He). The GENEPOP v.4.0.10 program [35] was used to test for Hardy–Weinberg Equilibrium (HWE) and Linkage Disequilibrium (LD), with 10,000 dememorizations, 1000 runs, and 10,000 comparisons per run. The Bonferroni correction was applied to the resulting HWE and LD matrices. The allelic richness (Ar) and the inbreeding coefficient (FIS) were calculated in FSTAT 2.9.3 [36]. The frequencies of the private alleles were estimated in GDA 1.1 [37]. Please note here that the preliminary analysis of the data from the populations of Governador Eugênio Barros, Timon, Floriano, Parnaíba, and Teresina, indicated that they formed a single cluster (denominated Piauí here), in which they are considered to be a single panmictic population by the STRUCTURE analysis. These samples were considered a single panmictic population, this criterion was used for AMOVA and Linkage Disequilibrium (LD) analyses.

The genetic differentiation of the populations was evaluated based on an Analysis of Molecular Variance (AMOVA), the FST index (genetic differentiation), and Nm (gene flow), all of which were calculated in ARLEQUIN 3.5 [38] with 10,000 permutations. Population structure was analyzed in STRUCTURE 2.3.4 [39], which is based on a Bayesian approach through a model that identifies the groups from their Hardy–Weinberg and linkage equilibria. The samples were assigned to a specific number of groups (K), with intervals of K = 1–12 and 20 repetitions, and a burn-in of 250,000 cycles followed by 750,000 Markov Chain Monte Carlo (MCMC) samples, assuming random sorting and correlated allele frequencies. The most probable K value was determined using the Delta K method of Evanno et al. [40] in STRUCTURE HARVESTER [41]. The CLUMPP v1.1.2 program [42] was used to summarize the results of the 20 permutations of the selected K value and provide the Q matrices based on the study populations. The results were plotted in STRUCTURE SELECTOR [43].

The effective size Ne(pop) of each population was estimated in NEESTIMATOR 1.3 [44], based on the Linkage Disequilibrium (LD) model. The BOTTLENECK 1.2.02 program [45] was used to find evidence of bottleneck events in the different populations. This program applied three different mutational models to the analysis of the data—the Infinite Alleles Model (IAM), the Stepwise Mutation Model (SMM), and the Twin-Phase Mutation Model (TPM). The heterozygosity rates under mutation-drift equilibrium for the TPM model assumed 95% of single-step mutations and 5% of multiple-step mutations, as recommended for microsatellite loci by Excoffier and Lischer [38]. The significance of the results was determined using the Wilcoxon test, as recommended in the manual, for less than 20 markers.

## 3. Results

Six microsatellite loci were genotyped in the 138 *Ae. aegypti* specimens collected from the nine populations surveyed in the Mid-North region of Brazil, producing a total of 828 genotypes and 32 alleles. Between one and nine alleles were recorded per locus per population, with the largest number (9) being recorded for locus B07 from Piauí, which was also the study area with the highest allelic richness (5.368). The smallest number of alleles (1) was identified for locus 3472 from Governador Eugênio Barros and locus H08 from Senador Alexandre Costa, with both these populations also having the lowest allelic richness (Table 3).

In terms of the estimates of the effective number of alleles Ne(allele), the highest frequency was recorded in the population of Barra do Corda (B07), while the lowest frequencies were registered in Governador Eugênio Barros (3472) and Senador Alexandre Costa, H08 (Table 3). A total of nine private alleles were found in populations—(Barra do Corda, São Domingos, Senador Alexandre Costa, Vargem Grande e Piaui (Table 3). While the samples from Piaui had the highest number of private alleles, Governador Eugênio Barros presented no private alleles whatsoever. The loci with the most populations with null alleles were B19 (found in five populations) and B07, for which null alleles were found in all populations (Table 3).

The Hardy–Weinberg Equilibrium (HWE) analysis was based on 36 tests, of which 15 (41.66%) indicated a significant deviation after applying the Bonferroni correction (*p* < 0.05), most (13) indicating heterozygote deficiency. The populations of Barra do Corda (3), and Piauí (4) had the highest numbers of loci in disequilibrium, whereas Senador Alexandre Costa presented the highest number of loci in equilibrium (Table 4).

The Analysis of Molecular Variance (AMOVA) revealed highly significant genetic variability when all the Mid-North populations were arranged in a single group, with 14.34% of the variation being found among populations and 85.66% within populations, with FST = 0.14340 (*p* = 0.00000 ± 0.00000). A second scenario was generated with the samples arranged in two groups, as indicated by the STRUCTURE analysis (Table 5), with one group formed by samples from Maranhão (Barra do Corda, São Domingos, Senador Alexandre Costa, and Vargem Grande) and the second group by two populations from Maranhão (Governador Eugênio Barros and Timon) and the three populations from Piauí (Floriano, Paranaíba, and Teresina) (Table 5).

The analysis found no significant genetic structuring between the groups, with FCT = 0.04357 (*p* = 0.00759 ± 0.00304), although significant genetic differentiation was found between samples in the same group, with FSC = 0.12144 (*p* = 0.00000 ± 0.00000). In both scenarios analyzed, the percentage of within-population genetic variation (85.66%; 84.03%) was much higher than that found between populations (14.34%) or among populations within groups (13.47%) (Table 5).

The estimates of genetic differentiation based on the pairwise FST and gene flow (Nm) values indicated highly significant genetic differentiation (*p* = 0.00000 ± 0.0000) in all the populations, with the highest fixation index (FST = 0.378) being found between the samples from Governador Eugênio Barros (GEB) and Vargem Grande which, in turn, had the lowest gene flow (Nm = 0.819). By contrast, the lowest genetic differentiation (FST = 0.013) was recorded between Timon and Teresina and, as expected, the highest gene flow (Nm = 35.328) (Table 6).

The general analysis of population structure based on the Bayesian approach revealed that the most probable number of genetic groups in the populations sampled was K = 2, indicating a mixed ancestry from two lineages (Figure 2). The first lineage, represented by the predominance of the red gene set, joined together four populations from Maranhão (Barra do Corda [BC], São Domingos [SD], Senador Alexandre Costa [SAC], and Vargem Grande [VG]). The second lineage, dominated by the green gene set, grouped two populations from Maranhão (Governador Eugênio Barros [GEB] and Timon [TI]), and three populations from Piauí, i.e., Floriano (FL), Parnaíba (PAR), and Teresina (TE).

Effective population size Ne(pop) was estimated based on the Linkage Disequilibrium (LD) model, with the lowest value being recorded for the populations from Maranhão (9.6) and the highest value (45.5) for the populations from Piauí. The heterozygosity tests were based on the three mutation models (IAM, TPM, and SMM), and the populations from both Maranhão and Piauí presented a significant excess of heterozygosity for all three of the models tested, which indicates that these populations passed through a recent bottleneck (Table 7).

## 4. Discussion

The loci analyzed in the present study presented levels of polymorphism similar to those reported for *Ae. aegypti* from other locations around the world. A total of 10 alleles were recorded here for the B07 locus, for example, in comparison with 13 alleles in the populations from São Paulo [46] and only six in those from Manaus [47]. Similarly, the six alleles found at locus 3472 in the present study are comparable with the eight alleles recorded by Paupy et al. [48] in northern Cameroon. Microevolution has significant implications in epidemiology. Rapid changes in allele frequencies may involve non-neutral characters, which may eventually influence the susceptibility of a given mosquito population to insecticides and chemical repellents [46].

The genetic composition of the *Ae. aegypti* populations of the Brazilian Mid-North revealed the existence of two lineages that are present in higher or lower proportions at the different localities, with mixed ancestry and significant genetic differentiation (Fst = 0.14340; *p* = 0.000) in the region. 

The Bayesian analysis revealed clear evidence of genetic sorting in the samples, with the geographically closest populations grouping in the same cluster, given their genotypic similarities. However, some geographically distant populations were also genetically similar and were grouped in the same lineage, as in the case of the Governador Eugênio Barros (GEB) and Piauí populations. Given the reduced dispersal capacity of *Ae. aegypti*, which may range from only 50 m to 800 m [49,50], the genetic similarity found between different localities may be due to the passive dispersal of the mosquito, mediated by human transportation and highway traffic, which may enhance the connectivity of the different populations [51,52,53]. 

The clustering pattern of the first lineage indicated a low genetic connectivity between the populations (Nm = 0.819–5.505). This is evident in the high indices of genetic differentiation (FST = 0.083–0.378), which indicate limited levels of gene flow between populations. Other similar studies involving mitochondrial markers have revealed high levels of genetic differentiation in *Ae. aegypti* populations in Maranhão [24], with FST values ranging from −0.012 to 0.813. The findings of the present study reconfirm the existence of the two genetic lineages found by Fraga et al. [23] and Sousa et al. [24] using mitochondrial markers.

Maranhão state straddles the transition zone between the Amazon and Cerrado biomes, an ecotone with a high diversity of ecosystems, species, and climatic conditions [54]. The state is also an important tourist destination and shipping hub, subject to intense movements of both passengers and cargo, which may favor the introduction of new genetic groups of *Ae. aegypti.*

The second lineage also presented a high level of genetic connectivity, reflecting a high level of gene flow among its populations (Nm = 6.050–35.328), which is also reflected in its low FST values (0.013–0.076). This high level of gene flow may reflect the passive dispersal of *Ae. aegypti* throughout the region through the interconnection of long-distance transportation routes [55]. The genetic similarity between Timon and the populations from Piauí (FST = 0.013–0.039) in lineage 2 is probably related to the geographic proximity of these localities, which are connected by highways and waterways. It thus seems likely that the Timon population was founded by two or more waves of colonization originating in Piauí (PI).

The effective population Ne(pop) sizes recorded in the present study were relatively small but are consistent with the values recorded by Maitra et al. [16] in 15 Brazilian *Ae. aegypti* populations, based on 12 microsatellite loci Mendonça et al. [47] recorded a similar scenario in four populations from Manaus, in central Brazilian Amazonia, while Olanratmanee et al. [56] also found this pattern in five villages in Thailand. The low Ne values are likely the result of demographic processes, such as biased sex ratios, fluctuations in population size, and/or the differential breeding success of certain individuals [57].

This pattern is likely the result of successive genetic bottlenecks and founder events occurring either during the colonization process or as a consequence of measures taken to control the local mosquito populations. These processes cause a reduction in effective population size Ne(pop), which may reinforce the differentiation of populations by genetic drift [16]. This was especially clear in the samples from Maranhão, which had the lowest Ne(pop) values and, in turn, the highest levels of genetic divergence among its populations.

The analysis of genetic bottlenecks indicated that the study populations are not undergoing a process of expansion. However, there is clear evidence of the effects of genetic bottlenecks in both the Maranhão and Piauí populations, with all three mutation models tested here indicating that these populations have undergone a drastic reduction in size in the past, which is consistent with the low Ne values recorded for the populations. Invasive species often go through population bottlenecks, which tend to reduce their genetic diversity drastically, and in particular their allelic richness [58].

The populations from Maranhão and Piauí presented distinct demographic patterns. While evidence of genetic bottlenecks was found in both cases, effective population sizes were very small in Maranhão in comparison with the much higher Ne values recorded in Piauí. The demographic dynamics of *Ae. aegypti* populations may be influenced by environmental, climatic, and social factors (including human population density), and the intensive application of insecticides, which may generate distinct patterns of genetic structure in the *Ae. aegypti* populations [47,59,60]. These factors may combine to differentiate the genetic makeup of the different localities through the successive expansion and reduction or extinction and recolonization of these populations [9]. 

This intra-population variability may be derived from differences in the vectorial capacity of the mosquito, its predisposition to the virus, or even its resistance to insecticides or other ecological adaptations, all of which may limit the effectiveness of control measures [23].

The understanding of the genetic structure of *Ae. aegypti* populations and their relatedness is fundamentally important for the development of effective programs of vector control, given that populations with a reduced effective size may be more subject to the effects of genetic drift, which may lead to an intense loss of genetic diversity over only a few generations. For programs that target the genetic diversity of the mosquito populations, the effective size of the target population must be taken into consideration to ensure the success of control measures [16]. The commitment of local human populations is also extremely important to ensure the success of any initiative to combat this potentially dangerous disease vector.

## 5. Conclusions

The findings of the present study reveal the coexistence of two distinct sets of genes in the *Ae. aegypti* populations of the Mid-North region of Brazil, based on the analysis of microsatellite markers. The results of the analysis indicate high levels of genetic differentiation in the Maranhão populations and marked genetic connectivity among the Piauí populations, which present low levels of genetic differentiation that indicate frequent gene flow, presumably mediated by passive dispersal. The results of this study also indicate the possible introduction of a new set of genes from Piauí into Maranhão. These data on the connectivity and genetic isolation of the different *Ae. aegypti* populations represent valuable input for the establishment of guidelines for the development of innovative strategies for the effective control of the populations of this disease vector.

## Figures and Tables

**Figure 1 insects-14-00530-f001:**
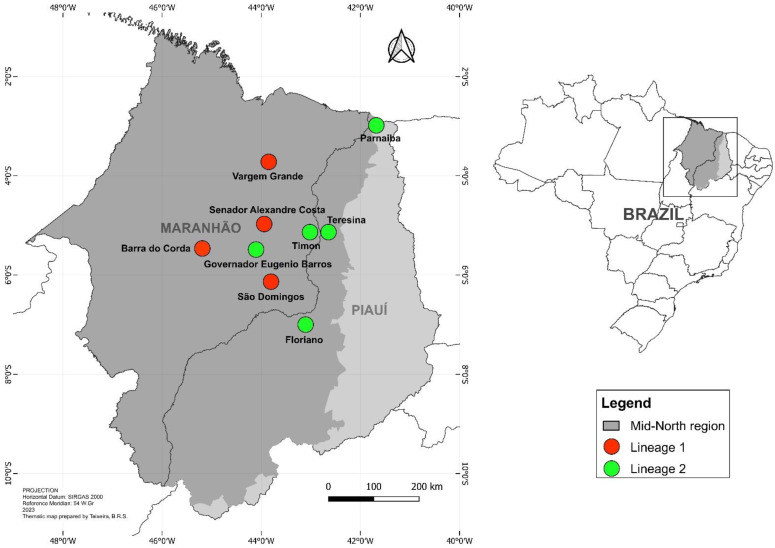
The municipalities with high Building Infestation Rates (BIRs) of *Ae. aegypti* that were surveyed in the Mid-North region (Maranhão and Piauí states) of northeastern Brazil. The color of each municipality indicates its predominant *Ae. aegypti* lineage, i.e., lineage 1 (red) or 2 (green).

**Figure 2 insects-14-00530-f002:**
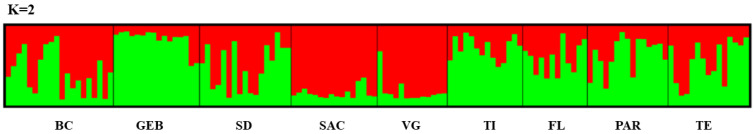
Genetic groups (K = 2) in the *Ae. aegypti* populations from the Mid-North region of Brazil sampled in the present study. Red areas = (BC: Barra do Corda; SD: São Domingos; SAC: Senador Alexandre Costa; VG: Vargem Grande); Green Areas = (GEB: Governador Eugênio Barros; TI: Timon; FL: Floriano; PAR: Parnaíba; TE: Teresina).

**Table 1 insects-14-00530-t001:** Geographic coordinates and sample sizes of the areas surveyed in the Mid-North region of northeastern Brazil.

State	Population	Abbreviation	Coordinates (Lat./Long.)	Sample Size
Maranhão	Barra do Corda	BC	5°30′21″ S	20
			45°14′6″ W	
	Governador Eugênio Barros	GEB *	5°18′39″ S	16
			44°14′51″ W	
	São Domingos	SD	5°34′46″ S	17
			44°22′59″ W	
	Senador Alexandre Costa	SAC	5°15′21″ S	16
			44°03′21″ W	
	Vargem Grande	VG	3°32′34″ S	13
			43°54′57″ W	
	Timon	TI *	5°05′38″ S	14
			42°50′13″ W	
Piauí	Floriano	FL	6°46′2″ S	12
			43°1′33″ W	
	Parnaíba	PAR	2°54′14″ S	15
			41°46′35″ W	
	Teresina	TE	5°05′20″ S	15
			42°48′07″ W	

* *Ae. aegypti* populations that were collected in the Maranhão state but that grouped in the cluster Piauí.

**Table 2 insects-14-00530-t002:** Description of the six microsatellite loci analyzed in *Ae. aegypti* in the present study.

*Locus*	Type of Repeat	F	pb	r (Median)	References
3472	GAAA(GA)_6_ CAGACAGGAAA	FAM	82–117	−0.0071	[26]
B19	CAT_7_	VIC	101–205	0.0970	[27]
B07	GA_15_	NED	95–195	0.1040	[27]
M201	ATA_36_	PET	92–140	−0.3894	[27]
H08	TCG_7_	VIC	101–218	−0.0097	[27]
T3A7	(TTTA)_7_(T)_14_	PET	107–263	0.0045	[26]

F = fluorophore; pb = base pairs; r = null allele frequency.

**Table 3 insects-14-00530-t003:** Genetic diversity of the samples of *Ae. aegypti* collected from nine municipalities in the Mid-North region of Brazil.

*Locus*		BC	GEB	SD	SAC	VG	PI
3472	N	20	16	17	16	13	56
	Na	4	1	3	4	2	6
	Ne(allele)	2.694	1.000	2.181	3.413	1.166	2.899
	Ar	3.952	1.000	3.000	4.000	2.000	4.669
	Ap	-	-	-	-	-	2
	r	0.244	-	-	-	0.238	-
B19	Na	4	3	5	3	2	3
	Ne(allele)	3.606	2.522	2.249	2.667	1.988	2.154
	Ar	4.000	3.000	4.665	3.000	2.000	2.858
	Ap	-	-	1	-	-	-
	r	0.300	0.392	-	0.195	0.035	0.017
B07	Na	5	3	5	3	5	9
	Ne(allele)	3.980	2.024	2.833	1.679	3.449	3.670
	Ar	4.952	3.000	4.705	2.970	5.000	5.368
	Ap	-	-	-	-	1	2
	r	0.193	0.039	0.189	0,075	0.176	0.126
M201	Na	2	2	2	3	2	2
	Ne(allele)	1.471	2.000	1.940	1.962	1.257	1.849
	Ar	2.000	2.000	2.000	2.813	2.000	2.000
	Ap	-	-	-	1	-	-
	r	0.029	-	-	-	-	-
H08	Na	3	2	3	1	2	3
	Ne(allele)	2.986	1.800	2.261	1.000	1.257	2.974
	Ar	3.000	2.000	2.986	1.000	2.000	3.000
	Ap	-	-	-	-	-	-
	r	-	-	0,281	-	-	0.122
T3A7	Na	4	2	2	3	2	4
	Ne(allele)	1.867	1.882	1.841	2.977	1.649	2.544
	Ar	3.651	2.000	2.000	3.000	2.000	3.411
	Ap	1	-	-	-	-	1
	r	0.064	0.402	-	-	0.011	0.003
Median	Na	3.666	2.166	3.333	2.833	2.500	4.500
	Ne(allele)	2.747	1.871	2.218	2.334	1.794	2.682
	Ar	3.593	2.167	3.226	2.797	2.500	3.551

BC: Barra do Corda; GEB: Governador Eugênio Barros; SD: São Domingos; SAC: Senador Alexandre Costa; VG: Vargem Grande; PI: (Timon; Floriano; Parnaíba; Teresina); N: number of analyzed mosquitoes per population; Na: number of alleles; Ne(allele): number of effective alleles; r: null allele frequency; Ar: allele richness; Ap: frequency of private alleles and (-): absence.

**Table 4 insects-14-00530-t004:** Measures of heterozygosity, the results of the tests of Hardy–Weinberg Equilibrium, and the inbreeding coefficients recorded for the populations of *Ae. aegypti* sampled in the present study.

*Locus*		BC	GEB	SD	SAC	VG	PI
3472	Ho	0.300	0.000	0.647	0.750	0.000	0.673
	He	0.644	0.000	0.542	0.707	0.142	0.655
	P-HWE	**0.001**	-	0.744	0.720	**0.040**	**0.022**
	FIS	0.541	NA	−0.166	−0.029	1.000	−0.018
B19	Ho	0.250	0.063	0.529	0.357	0.462	0.509
	He	0.713	0.604	0.555	0.630	0.497	0.536
	P-HWE	**0.000**	**0.000**	**0.000**	**0.006**	0.560	0.085
	FIS	0.668	0.903	0.077	0.463	0.111	0.059
B07	Ho	0.450	0.438	0.412	0.438	0.462	0.537
	He	0.749	0.506	0.647	0.486	0.710	0.728
	P-HWE	**0.000**	**0.027**	0.079	0.370	**0.021**	**0.008**
	FIS	0.420	0.167	0.390	0.132	0.385	0.271
M201	Ho	0.320	1.000	0.824	0.750	0.231	0.607
	He	0.328	0.500	0.484	0.490	0.204	0.459
	P-HWE	0.578	1.000	1.000	1.000	1.000	0.997
	FIS	0.088	−1.000	−0.684	−0.506	−0.091	−0.314
H08	Ho	1.000	0.667	0.200	0.000	0.231	0.500
	He	0.665	0.444	0.558	0.000	0.204	0.664
	P-HWE	1.000	1.000	**0.001**	-	1.000	**0.007**
	FIS	−0.482	−0.474	−0.684	NA	−0.091	0.256
T3A7	Ho	0.389	0.000	0.706	0.750	0.385	0.500
	He	0.465	0.469	0.457	0.664	0.393	0.607
	P-HWE	0.183	**0.000**	1.000	0.591	0.667	**0.001**
	FIS	0.190	1.000	−0.524	−0.098	0.063	0.185
**Median**	Ho	0.448	0.361	0.553	0.507	0.295	0.519
	He	0.590	0.420	0.540	0.496	0.358	0.566
	FIS	0.237	0.099	−0.041	−0.006	0.000	0.073

BC: Barra do Corda; GEB: Governador Eugênio Barros; SD: São Domingos; SAC: Senador Alexandre Costa; VG: Vargem Grande; PI: (Timon; Floriano; Parnaíba; Teresina); Ho: observed heterozygosity; See per *Locus*; He: expected heterozygosity; NA: Not Analyzed; FIS: inbreeding coefficient; P-HWE: the values in bold type indicate disequilibrium (Hardy–Weinberg) at the respective localities, following the Bonferroni sequential correction (*p* < 0.0042).

**Table 5 insects-14-00530-t005:** Results of the AMOVA for the *Ae. aegypti* populations sampled from the Mid-North region of Brazil.

Groups of Samples	Source of Variation	Variation (%)	Fixation Index
All: BC, GEB, SD, SAC, VG, TI, FL, PAR, and TE	Among population	14.34	FST: 0.14340 ***
	Within populations	85.66	
Two Groups: Maranhão: BC, SD, SAC and VG Piauí: GEB, TI, FL, PAR, and TE	Among groups	4.36	FST: 0.15971 ***
	Among populations within groups	13.47	FSC: 0.12144 ***
	Within populations	85.01	FCT: 0.04357

BC: Barra do Corda; GEB: Governador Eugênio Barros; SD: São Domingos; SAC: Senador Alexandre Costa; VG: Vargem Grande; TI: Timon; FL: Floriano; PAR: Parnaíba; TE: Teresina. Significance was tested by 10,000 permutations. The fixation index (FST) within the samples. The fixation index (FCT) between regions. The fixation index (FSC) among samples between the regions. *** *p* = 0.00000 ± 0.00000.

**Table 6 insects-14-00530-t006:** Genetic differentiation, gene flow, and the approximate geographic distances between the *Ae. aegypti* populations sampled in the present study.

POP	BC	GEB	SD	SAC	VG	TI	FL	PAR	TE
BC		2.406	5.505	2.215	4.020	5.909	5.591	4.990	12.239
GEB	0.172 (130.6)		4.331	1.163	0.819	4.079	2.382	5.419	3.168
SD	0.083 (132.9)	0.103 (68.5)		2.369	1.463	4.372	4.339	3.737	7.277
SAC	0.184 (163)	0.300 (38.8)	0.174 (100.9)		1.353	1.705	1.842	2.440	2.362
VG	0.110 (322)	0.378 (241.6)	0.254 (265)	0.269 (280.3)		1.357	1.655	1.366	2.277
TI	0.078 (331.9)	0.109 (207.8)	0.102 (245.4)	0.226 (169.7)	0.269 (305.7)		12.317	15.054	35.328
FL	0.0820 (388.9)	0.173 (324.5)	0.103 (256.6)	0.213 (330.3)	0.231 (555.2)	0.039 (249.7)		6.050	17.486
PAR	0.091 (641.4)	0.084 (517.3)	0.117 (554.9)	0.170 (512.3)	0.267 (318.2)	0.032 (343.5)	0.076 (572.6)		8.141
TE	0.039 (342.8)	0.136 (212.3)	0.064 (256.3)	0.174 (180.6)	0.180 (316.6)	0.013 (13.8)	0.027 (255.5)	0.057 (335.4)	

POP: Populations; BC: Barra do Corda; GEB: Governador Eugênio Barros; SD: São Domingos; SAC: Senador Alexandre Costa; VG: Vargem Grande; TI: Timon; FL: Floriano; PAR: Parnaíba; TE: Teresina. (FST—below the diagonal), (Nm—above the diagonal), (km—within parentheses). All the FST and Nm values are highly significant (*p* = 0.0000 ± 0.0000).

**Table 7 insects-14-00530-t007:** Effective population sizes Ne(pop) and the results of the bottleneck tests for the *Ae. aegypti* populations sampled in the presented study in the Mid-North region of Brazil.

Model		Maranhão	Piauí
LD		9.6	45.5
95% CI		4.1–16.3	19.6–193.4
IAM	He < Heq	0	0
	He > Heq	6	6
	P (He > Heq)	**0.0078**	**0.0078**
TPM	He < Heq	0	1
	He > Heq	6	5
	P (He > Heq)	**0.0078**	**0.0234**
SMM	He < Heq	2	2
	He > Heq	4	4
	P (He > Heq)	**0.2812**	**0.5781**

Maranhão: (Barra do Corda; São Domingos; Senador Alexandre Costa and Vargem Grande); Piauí: (Governador Eugênio Barros, Timon; Floriano; Parnaíba and Teresina); LD: Linkage Disequilibrium; CI: Confidence Interval; IAM: Infinite Alleles Model; SMM: Stepwise Mutation Model; TPM: Twin-Phase Mutation Model; He: expected heterozygosity; Heq: heterozygosity expected under mutation and drift equilibrium; He > Heq: number of loci with an excess of heterozygotes; P (He > Heq): Probability of an excess of heterozygotes at the locus based on the results of the Wilcoxon test. The values highlighted in bold script indicate a significant bottleneck (*p* < 0.05).

## Data Availability

The database used for this study is available https://www.doi.org/10.6084/m9.figshare.23295272.

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
