# Peer review of "Genetic Differentiation of Aedes aegypti (Diptera: Culicidae) in Areas with High Rates of Infestation in Mid-North Region of Brazil"

_insects, 2023, doi:10.3390/insects14060530_

Round 1

Reviewer 1 Report

since you start  with affirmation of high levels of genetic differentiation, which represent a potential risk for 23 the occurrence of epidemics and a major problem for local public health authorities. such an important statement, just wonder if other research group  both in Brazil and out of Brazil get to same collusion? I am wonder why  there is no mention about vector competence in reared lab Ae aegypti and also with wild species?

Author Response

Response to Reviewer 1 Comments

Point 1 - Since you start with affirmation of high levels of genetic differentiation, which represent a potential risk for the occurrence of epidemics and a major problem for local public health authorities. Such an important statement, just wonder if other research group both in Brazil and out of Brazil get to same collusion? I am wonder why there is no mention about vector competence in reared lab Ae aegypti and also with wild species?

Answer - In the manuscript there are several mentions about the relationship between the genetic variability of Ae. aegypti and the competence of wild species. For example: see, Ln 65-72.

In relation the vectorial competence in reared lab Ae. aegypti, the present study does not focus on this group, because we are working with wild species.

Reviewer 2 Report

Overall this is a solid piece of work. My main comments are to suggest some changes to the intro to make the goals more explicit. In addition, I encourage you to double-check the methods and results for the Fst analysis, your results are in pretty strong contrast to many other studies and this should be 1. confirmed, 2. discussed.

Ln 75 

The weakness of SSRs is that homology can be hard to determine because the sequence is short and its genomic location is unknown. I think that a set of known-sequence probes—like a SNP array—would be more informative. Or high-throughput short read sequencing. I understand that there are economic reasons to use SSRs and that they are sufficient, but it is inaccurate to describe SSRs as “ideal.” Better to be transparent about the reasons they are the best choice FOR THIS STUDY.

Ln 83

You have not yet made it clear that there is a direct relationship between high levels of genetic diversity in A. aegypti and an increased likelihood of disease transmission.

Ln 89

From this, the point seems to be that: if one wishes to use a genetic control measure, the genetic variability of the mosquito needs to be assessed. Up until this point, I assumed you were arguing that variation in mosquito genotypes raises the probability that a genotype that is capable of vectoring novel forms of the listed diseases will arise...

Ln 200

You should explore other ways of correcting for multiple testing. Permutation to obtain empirical significance thresholds is generally preferred to Bonferroni.

Ln 225

these numbers seem quite extreme. looking at several other papers, the level of within-population divergence you report is surprising to me. I am not an expert on A. leg and defer to the authors, but I encourage them to look over recent literature to be sure that their methods and manner of reporting are directly comparable to similar studies.

Author Response

Response to Reviewer 2 Comments

Overall this is a solid piece of work. My main comments are to suggest some changes to the intro to make the goals more explicit. In addition, I encourage you to double-check the methods and results for the Fst analysis, your results are in pretty strong contrast to many other studies and this should be 1. confirmed, 2. discussed.

Ln 75

The weakness of SSRs is that homology can be hard to determine because the sequence is short and its genomic location is unknown. I think that a set of known-sequence probes—like a SNP array—would be more informative. Or high-throughput short read sequencing. I understand that there are economic reasons to use SSRs and that they are sufficient, but it is inaccurate to describe SSRs as “ideal.” Better to be transparent about the reasons they are the best choice FOR THIS STUDY.

Response - The SSRs are good molecular markers for the study of population genetics, given their high rates of mutation and polymorphism [12,13]. In particular, SSRs are low-cost, neutral markers which, in recent years, have provided valuable insights into genetic diversity, gene flow patterns, and intra- and inter-specific genetic differentiation in a range of studies, both in Brazil and around the worlde [14-17]. See lines 60-64

Ln 83

You have not yet made it clear that there is a direct relationship between high levels of genetic diversity in A. aegypti and an increased likelihood of disease transmission.

Response - Studies of the genetic variability of Ae. aegypti populations are extremely important for the Mid-North region of Brazil, where a large proportion of the municipalities are at risk of infestation by this vector and are thus faced with the potential spread of the dengue, chikungunya, and zika arboviroses [10,18-22]. In two previous studies in Ma-ranhão, one of the Brazilian states that make up the Mid-North region, Fraga et al. [23] and Sousa et al. [24] found high levels of intra-population genetic differentiation in Ae. aegypti, with evidence of the coexistence of two distinct genetic lineages in the state. It is important to note that populations of vectors with high levels of genetic differenti-ation within a given (micro or macro) region may present a greater competence for the transmission of diseases. See lines 65-72.

Ln 89

From this, the point seems to be that: if one wishes to use a genetic control measure, the genetic variability of the mosquito needs to be assessed. Up until this point, I assumed you were arguing that variation in mosquito genotypes raises the probability that a genotype that is capable of vectoring novel forms of the listed diseases will arise.

Response - Based on several studies carried out that showed the importance of population genetics to demonstrate the competence of vectors in the transmission of viruses, as well as in the emergence of new viruses, for example the zika virus, which is more recente. See lines 49-57.

Ln 200

Answer - You should explore other ways of correcting for multiple testing. Permutation to obtain empirical significance thresholds is generally preferred to Bonferroni.

Response - The Bonferroni correction was chosen because it is the most widely adopted multiple test correction test in Ae. aegypti population genetics work.

Ln 225

these numbers seem quite extreme. looking at several other papers, the level of within-population divergence you report is surprising to me. I am not an expert on A. leg and defer to the authors, but I encourage them to look over recent literature to be sure that their methods and manner of reporting are directly comparable to similar studies.

Response – Our data are consistent and similar to the results found by Maitra et al, 2019 when studying Brazilian populations of Ae. aegypti who found the Analysis of Molecular Variance (AMOVA) revealed highly significant genetic variability when all the Mid-North populations are arranged in a single group, with 10.72% of the variation being found among populations and 89.28% within populations, with FST = 0.1072.

Reviewer 3 Report

Dr. Farias Rodrigues and Colleagues have genotyped 138 individuals of A. aegypti, from 9 locations in northeastern Brazil covering an area of 400x400Km, at 6 microsatellite loci. They study variability, connectivity and investigate the presence/distribution of genetic groups in the area.

The study is not overly rich in terms of data, but the fact that it targets a species of utmost interest, as well as the fact that the area investigated may not be easily accessible to researchers worldwide, make the study valuable. Pending some necessary modifications to the analysis and presentation (see below) my auspice is that it will eventually be published.

Major

In the global Structure analysis two groups may be identified, although the level of mixing is as high as to make it not really easy to define. If nevertheless the discussion, and identification of ‘cluster 1’ and ‘cluster 2’, is around populations that are mostly red and populations that are mostly green, I would say that population 2 (Eugenio Barros) is green, and as such should be identified as part of cluster 2. Note that this division among clusters 1 and 2 (and the accompanying identification of populations as belonging to one or the other) has many implications throughout the text (description) and analysis (second ANOVA). If my interpretation that the structure analysis is the foremost source of the identification of the two groups is not correct, please clarify. Otherwise correct presentation and repeat analyses with a correct assignment of locations/populations to clusters.

The statement at line 303/304, or high differentiation between 5 lineages within the two aforementioned clusters apparently does not find support in the second structure analysis. While some substructure is visible within cluster 1 (limited to populations 2-4-5), no substructure altogether is seen in Cluster 2. The statement at line 312 about a new gene pool needs clarification and possibly support. Can the three subclusters in cluster 1 be identified as 2 corresponding to groups previously reported in the literature and 1 as a new cluster? Or the only support for the existence of a new previously unreported lineage is that the Evanno method identified 3 as the optimal number of clusters?

The description and interpretation of linkage disequilibrium is not very clear. What evidence do the authors have? I assume it is not a matter of physical linkage of loci on the chromosomes, as 6 loci in a genome can hardly be linked. Which is their interpretation of the results? Do they think LD is an unwanted spurious effect of the limited number of loci/individuals, and as such a ‘nuisance’, or the outcome of some biological process they wish to discuss? Lines 338-342 are fairly vague in this respect.

Minor:

lines 26/27 ‘different sets of genes’. The authors repeatedly use the term ‘sets of genes’ in different contexts throughout the paper. The actual meaning of this term is not clear to me. If this is intended of a synonym of ‘populations’, please use this latter term. Otherwise, define the sentence.

34/35 ‘determining the genotypic components ….’

40 and 176 ‘one to seven per locus’ specify ‘per population’

70 ‘given the epidemiological…’ I would state this differently. The epidemiological importance of the species calls for results on its population structure. The choice on the markers to use should be based on the likely informativeness of different types of markers for the problem at hand.

Figure 1 should be redrawn. It looks like the direct output of some geographic information system. I would use this as the base for redrawing. The second small panel on the right seems useless. The use of symbols in picture and key is equally useless, as they are all triangles. The meaning of the hatching on the main map is not indicated.

108 if some caution was taken to avoid genotyping individuals from a single pallet/egg mass/family, please specify, as this would be a positive aspect of sampling.

114 ‘microbes’?

117 ‘isolated’? I understand these loci come from the literature, here possibly ‘amplified’.

121 a different amount of forward and reverse primer was used, please confirm.

122 it appears that the fluorescent dye was mixed in the PCR and not physically conjugated to one primer. This method is not familiar to me, please check.

Table 2 ‘CAT’ in locus B19 does not have an indication of the repeat number in the reference allele.

Table 2 terms ‘F, pb and r(Median)’ should be specified in the caption.

140/141 the difference between ‘null alleles’ and ‘loss of alleles’ is not clear to me. Is there a difference of they are used and synonyms?

143 and 163 it appears that two different procedures were used to estimate Ne. How are they different? Which one was used in the subsequent description? If only one is reported, the calculation of the other may be removed for clarity.

160 ‘interactions’?

Table 3 please define ‘r’ in the caption as well.

Table 4 In Ho lines, which is the difference from numeric zero and the keyword ‘mono’?

Table 4 I do not see the cited ‘values in bold’.

Table 4 Insert ‘see XXX for locus codes’ or repeat them.

223 ‘genetic variability’. Since ANOVA is about partitioning genetic variability across levels, please be clear about what type of ‘genetic variability’ you are referring to. Is this ‘between populations/sampling sites’?

Table 5 first and last column are difficult to read as they do not readily align with lines. I assume column one has to be independent from lines and last column has to be in line. Please reformat.

238-242 Values reported in the text appear to be different form what is reported in the table. Between groups=FCT?. Please revise carefully.

Table 6, caption. Having p-values for all FST and Nm all at zero seems not to be in line with the generally low structuring observed in the structure analysis. Please comment. Furthermore, caption appears to be truncated.

Figure 2 please indicate locations names or codes in the figure, as numeric codes are not easy to parse. Please improve the caption.

317 the situation in cluster 2 is generally different from cluster 1, is the ‘also’ lexical linker appropriate to introduce the sentence?

Throughout:

Location abbreviations. These have to be consistent throughout text, figures and tables. I noticed that population Floriano is not consistent. Please check others as well.

The name of the species has to be italicized, even in figure/table captions.

Generally appropriate. Some rewording of technical statements needed.

Author Response

Response to Reviewer 3 Comments

Dr. Farias Rodrigues and Colleagues have genotyped 138 individuals of A. aegypti, from 9 locations in northeastern Brazil covering an area of 400x400Km, at 6 microsatellite loci. They study variability, connectivity and investigate the presence/distribution of genetic groups in the area.

The study is not overly rich in terms of data, but the fact that it targets a species of utmost interest, as well as the fact that the area investigated may not be easily accessible to researchers worldwide, make the study valuable. Pending some necessary modifications to the analysis and presentation (see below) my auspice is that it will eventually be published.

Major

In the global Structure analysis two groups may be identified, although the level of mixing is as high as to make it not really easy to define. If nevertheless the discussion, and identification of ‘cluster 1’ and ‘cluster 2’, is around populations that are mostly red and populations that are mostly green, I would say that population 2 (Eugenio Barros) is green, and as such should be identified as part of cluster 2. Note that this division among clusters 1 and 2 (and the accompanying identification of populations as belonging to one or the other) has many implications throughout the text (description) and analysis (second ANOVA). If my interpretation that the structure analysis is the foremost source of the identification of the two groups is not correct, please clarify. Otherwise correct presentation and repeat analyses with a correct assignment of locations/populations to clusters.

Answer - We repeat analyses with a correct assignment of locations/populations to clusters, see Ln 176-183, Table 5 and Ln 201-205, Figure 2.

The statement at line 303/304, or high differentiation between 5 lineages within the two aforementioned clusters apparently does not find support in the second structure analysis. While some substructure is visible within cluster 1 (limited to populations 2-4-5), no substructure altogether is seen in Cluster 2. The statement at line 312 about a new gene pool needs clarification and possibly support. Can the three subclusters in cluster 1 be identified as 2 corresponding to groups previously reported in the literature and 1 as a new cluster? Or the only support for the existence of a new previously unreported lineage is that the Evanno method identified 3 as the optimal number of clusters?

Answer - We repeat analyses with the Evanno method identified 2 as the optimal number of clusters (k=2), see, Figure 2.

The description and interpretation of linkage disequilibrium is not very clear. What evidence do the authors have? I assume it is not a matter of physical linkage of loci on the chromosomes, as 6 loci in a genome can hardly be linked. Which is their interpretation of the results? Do they think LD is an unwanted spurious effect of the limited number of loci/individuals, and as such a ‘nuisance’, or the outcome of some biological process they wish to discuss? Lines 338-342 are fairly vague in this respect.

Answer - Based on the reviewer's analysis, we have heeded the guidance and removed and paragraph dealing with the description of the link imbalance in lines 338-342.

Minor:

lines 26/27 ‘different sets of genes’. The authors repeatedly use the term ‘sets of genes’ in different contexts throughout the paper. The actual meaning of this term is not clear to me. If this is intended of a synonym of ‘populations’, please use this latter term. Otherwise, define the sentence.

Answer – line 23 different genetic lineages

34/35 ‘determining the genotypic componentes - Answer - population structure, see line 31.

40 and 176 ‘one to seven per locus’ specify ‘per population’ – Ajusted, see line 35 and line 150.

70 ‘given the epidemiological…’ I would state this differently. The epidemiological importance of the species calls for results on its population structure. The choice on the markers to use should be based on the likely informativeness of different types of markers for the problem at hand.

Answer – Got corrected

Figure 1 should be redrawn. It looks like the direct output of some geographic information system. I would use this as the base for redrawing. The second small panel on the right seems useless. The use of symbols in picture and key is equally useless, as they are all triangles. The meaning of the hatching on the main map is not indicated.

Answer – The Figure 1 should was redrawn.

108 if some caution was taken to avoid genotyping individuals from a single pallet/egg mass/family, please specify, as this would be a positive aspect of sampling.

Answer - The some caution was taken to avoid genotyping individuals brothers, see lines 97-104.

114 ‘microbes’? Answer – microtubes

117 ‘isolated’? I understand these loci come from the literature, here possibly ‘amplified’.

Answer – Got corrected

121 a different amount of forward and reverse primer was used, please confirm.

Answer – Yes

122 it appears that the fluorescent dye was mixed in the PCR and not physically conjugated to one primer. This method is not familiar to me, please check.

Answer - The amounts of forwad and reverse primers are different because the complement of the forwad primer volume will be the fluorescent dye that was synthesized separately, a strategy that allows the use of this fluorescent dye in other works carried out by the research group.

Table 2 ‘CAT’ in locus B19 does not have an indication of the repeat number in the reference allele.

Answer – Got corrected

Table 2 terms ‘F, pb and r(Median)’ should be specified in the caption.

Answer – Got corrected

140/141 the difference between ‘null alleles’ and ‘loss of alleles’ is not clear to me. Is there a difference of they are used and synonyms?

Answer – They are synonyms, got corrected

143 and 163 it appears that two different procedures were used to estimate Ne. How are they different? Which one was used in the subsequent description? If only one is reported, the calculation of the other may be removed for clarity. Answer – They are different in that one analysis estimates the effective number of alleles in a population and the other analysis estimates the effective population size. Both should be kept, subtitle has been corrected.

It was effective allele frequency analysis.

160 ‘interactions’? Answer – Got corrected

Table 3 please define ‘r’ in the caption as well. Answer – Got corrected

Table 4 In Ho lines, which is the difference from numeric zero and the keyword ‘mono’?

Answer – Got corrected

Table 4 I do not see the cited ‘values in bold’.

Answer – Got corrected

Table 4 Insert ‘see XXX for locus codes’ or repeat them.

223 ‘genetic variability’. Since ANOVA is about partitioning genetic variability across levels, please be clear about what type of ‘genetic variability’ you are referring to. Is this ‘between populations/sampling sites’?

Answer - The levels were presented in the text, see lines 176-181.

Table 5 first and last column are difficult to read as they do not readily align with lines. I assume column one has to be independent from lines and last column has to be in line. Please reformat.

Answer – Got corrected

238-242 Values reported in the text appear to be different form what is reported in the table. Between groups=FCT?. Please revise carefully.

Answer – Got corrected

Table 6, caption. Having p-values for all FST and Nm all at zero seems not to be in line with the generally low structuring observed in the structure analysis. Please comment. Furthermore, caption appears to be truncated.

Answer – Got corrected

Figure 2 please indicate locations names or codes in the figure, as numeric codes are not easy to parse. Please improve the caption.

Answer – Got corrected

317 the situation in cluster 2 is generally different from cluster 1, is the ‘also’ lexical linker appropriate to introduce the sentence?

Answer – Got corrected

Throughout:

Location abbreviations. These have to be consistent throughout text, figures and tables. I noticed that population Floriano is not consistent. Please check others as well.

Answer – Got corrected

Reviewer 4 Report

Rodrigues et al have done a nice microsat study of the Aedes aegypti population in the Mid-North region of Brazil. They have done the right analyses, and the study is a valuable contribution to our knowledge of a medically important organism. However, I have some major concerns around the details of some of their analyses; in particular, the design of their STRUCTURE analysis needs to be reworked, and the population genetic analyses should be restructured to take into out whatever their new STRUCTURE results are. More detailed feedback follows:

MAJOR THINGS: All of these need to be addressed.

106-114: How many ovitraps were positioned per site? Is there a single ovitrap per municipality? Multiples? If multiple, how many, and how did you choose which individuals to select from which ovitrap? If there is a single ovitrap per municipality, I am a bit worried about whether your samples from each site are siblings--if you just got a bunch of siblings for each site, that is going to make sites look much more distinct than they actually are. If so, you might want to screen for likely siblings and remove them from the data set.  

266-276: I don't like the way the authors handle their STRUCTURE analysis. 

First bad STRUCTURE thing:

It seems wrong to me to run STRUCTURE, use it to identify populations, and then run STRUCTURE again on the sub-populations. The problem with this is that the authors say that this gives them 5 different genetic lineages (K=3 in one "cluster", and K=2 in another). But since they have done two different STRUCTURE  runs, there is no way of knowing whether this is true. Perhaps the two "sub-groups" in "cluster 2" are also found in "cluster 1", for a total of 3 different genetic lineages?

If you want to look at subgroupings within your two clusters, I think the appropriate thing to do is to look at the results from your other K=X runs. If there really are 5 "sub-groups" in your data set, then you should see this in the K=5 run and you can present those results. If you don't see that in the K=5 run, I think that reflects that your "sub-groups" are just artefacts of the weird tiered STRUCTURE design you have come up with. 

I am almost certain that you will find that your 5 sub-groups are artefactual based on the results for Cluster 2: when you run STRUCTURE on just those individuals, you get two clusters where every individual has about half its genetic background from each cluster. This is the result you get when you force-feed a single panmictic population into STRUCTURE: STRUCTURE "analyzes" the population by just splitting it right down the middle like that, and the Evanno method chooses k=2 because the Evanno method is incapable of evaluating (the correct) k=1. So I don't think you actually have any population structuring within Cluster 2 (at least, not that STRUCTURE is able to detect). 

Finally, I would recommend that you give us the results of the Evanno analysis, especially if you are going to ignore it and explore K=3, K=4, K=5, etc. That's not necessarily a bad thing to do (the Evanno method is mostly a rule of thumb), but the results of the Evanno analysis will help us figure out how valuable this is. If the delta-ln-whatever values for K=2, K=3, K=4, etc are all very similar, then that will support your desire to look at a bunch of different K-values. But if the value for K=2 is by far the best, that will be more evidence that there isn't much structure in the population apart from the two clusters you have identified. 

Second bad STRUCTURE thing:

STRUCTURE is coming too late in this analysis. You are using STRUCTURE to identify a bunch of "clusters" that have lots of gene flow within them but are genetically distinct from other clusters. This is a great use of STRUCTURE. However, this should happen BEFORE you do the rest of population genetic analysis. I.e., if FLOR/TE/PAR/TI represent a single panmictic population, it doesn't make sense to calculate Na, Ap, Ho, FIS, etc, for each one of those sampling sites separately; instead you should combine the individuals into a single "Piaui" population and calculate your pop-gen stats using all the individuals together. This will have the additional benefit of making Tables 3 and 4 much smaller (since you'll only have 2-3 populations) and easier to read and interpret. (If you want to keep the original tables with the site-by-site stats in the SI, I'd be OK with that).

Third bad STRUCTURE thing:

Speaking of identifying clusters: First, I recommend that you not use the term "clusters". This is confusing because you mean clusters-of-populations, but STRUCTURE is a clustering algorithm and the results that it produces are also "clusters" (i.e., all the red in Figure 2 is a "cluster"). You don't use "cluster" in this latter sense, but people who are familiar with STRUCTURE, etc, do, and so it makes things confusing. 

I recommend that you change your terminology like this: BC, SD, GEB, etc, should be called "sites". Then you can divide those "sites" up into "populations" (what you are now calling "clusters") based on the results of your STRUCTURE analysis. I think that this terminology does a better job of capturing the genetic reality that we are discussing, so I am ging to use it for the rest of this comment.

You never talk about how you divide the sites into clusters, and you imply that this division was an output of STRUCTURE. This simply isn't true; STRUCTURE doesn't cluster individuals or sites into populations (or at least, the mode of STRUCTURE you ran doesn't do that). Instead, defining populations is a bit of an art in which STRUCTURE data is useful, but you are still making qualitative judgements that you need to describe and justify to the reader. For instance, here is how I would do it: "STRUCTURE identified two genetic lineages; based on these results, we divided our sampling sites into two populations: the West population included those sites in which the first of the two lineages predominated (the red bars in Figure 2), i.e., BC, SD, SAC, and VG. The East population included those 

sites in which the second of the two lineages predominated (the green bars in Figure 2), i.e., GEB, TI, TE, PAR, and FLOR".

The weird thing here is that GEB looks like a Piaui population that has very little red heritage, even though it is right between two populations that are dominated by red. Regardless of whether you want to put GEB in a population with the other Maranhao sites or with the Piaui sites, I think you need to try to explain what is going on here: is there some geographic feature like a river that might be separating GEB from SD and SAC?

MODERATE THINGS: Most of these should be changed or addressed.

50: I believe the authority for Ae. aegypti is (Linnaeus, 1762), no italics. Linnaeus put the species in Culex, hence the parentheses.

76-91: I think it would be worth describing how your study differs from the previous studies in this region, either here or in the discussion. Are you just replicating their studies? Do you have more samples, more markers?

143, 163: It is awkward that you have two Ne variables. Can you rename them Ne(allele) and Ne(pop) or something like that?

176: I think you mean "per locus, per population", correct? You had more than 1 allele across all populations for all six of your microsats.

213-216: You mention some results here, but the data are never presented anywhere that I could find.

241-242: I don't think that the numbers in parentheses are correct: surely both 85%s should be in the first parentheses, and the second parentheses should be the ~14% numbers?

Table 2: This needs a caption describing what F, pb, etc mean.

Table 3: It would be nice to know the total number of alleles found across all 9 populations. Also, there is no explanation of what "r" means in the caption. Finally, the "Na" values you give at the bottom across all 6 microsats: you have calculated them as means, not medians (and perhaps done the same for Ne and Ar, though I didn't check).

Table 4: I recommend you make it easier to find the significant values in this table: maybe bold them? 

Table 6: There is a value in the BC/BC cell: what is that? Also, there are two values in the TI/TE and FLOR/TE cells above the diagonal--is this just a problem with the typesetting of the table, or are there actually two different values there?

Table 7: You might want to clarify in the caption whether these are the 6 Maranhao sites vs the 3 Piaui sites, or whether Maranhao means "Maranhao minus Timon", etc.

Figure 1: I think that the text in this figure should be in English (or in many cases it can probably be removed altogether). Why are there two insets of Brazil? I would delete the lower of the two; this would give you more space to make the key easier to read. I would encourage you to use different symbols for the two clusters you find; then it is easy for the reader to visualize how the TI site groups with the Piaui sites. Also, you refer to Piaui and Maranhao several times in the text, I think it would be valuable to put those names on the map here for the sake of your non-Brazilian readers (or your Brazilian readers who are bad at geography). Finally, I recommend that you explain in the caption that Mid-North is the shaded region; it's in the key, but not entirely obvious to me.

Figure 2: You need to include what numbers 6-9 stand for.

MINOR THINGS: I think these would improve the ms, but if you don't want to do them, that's fine.

32: Maybe "four very important arboviruses"? There are a lot more than these that it also vectors, no?

34: I would write "control measures"

53: You can lose the abbreviations (ZIKV, etc) as you never use them later on in the paper (you do the correct thing in spelling out the diseases later on).

94: this "region is a sub-region of Northeastern Brazil" sounds strange. Maybe just delete the "sub-" part?

107: "4-7 days"?

108: I'm not sure "pallets" is colloquial English, although without seeing your ovitraps I'm not sure what term to use--if they lay eggs onto paper, I might just call them "papers".

114: "microtubes"?

117: I would say "manufacturer's instructions" (like you do later in the paragraph).

140: You use "verify" in several places to mean "test for" or "analyze" or similar. I think you should restrict "verify" to mean "confirm something that we already believed to be true", which I don't think is what you mean here.

156: I think you mean "repetitions" or something like that, not "interactions".

Table 2: I would say "Repeat type"

Looks good; I noted above a couple of areas where the English didn't sound idiomatic (at least not to my American ear).

Author Response

Response to Reviewer 4 Comments

Rodrigues et al have done a nice microsat study of the Aedes aegypti population in the Mid-North region of Brazil. They have done the right analyses, and the study is a valuable contribution to our knowledge of a medically important organism. However, I have some major concerns around the details of some of their analyses; in particular, the design of their STRUCTURE analysis needs to be reworked, and the population genetic analyses should be restructured to take into out whatever their new STRUCTURE results are. More detailed feedback follows:

MAJOR THINGS: All of these need to be addressed.

106-114: How many ovitraps were positioned per site? Is there a single ovitrap per municipality? Multiples? If multiple, how many, and how did you choose which individuals to select from which ovitrap? If there is a single ovitrap per municipality, I am a bit worried about whether your samples from each site are siblings--if you just got a bunch of siblings for each site, that is going to make sites look much more distinct than they actually are. If so, you might want to screen for likely siblings and remove them from the data set.

Answer – This step was better detailed, see lines 96-104.

266-276: I don't like the way the authors handle their STRUCTURE analysis.

Answer – Got corrected

First bad STRUCTURE thing:

It seems wrong to me to run STRUCTURE, use it to identify populations, and then run STRUCTURE again on the sub-populations. The problem with this is that the authors say that this gives them 5 different genetic lineages (K=3 in one "cluster", and K=2 in another). But since they have done two different STRUCTURE runs, there is no way of knowing whether this is true. Perhaps the two "sub-groups" in "cluster 2" are also found in "cluster 1", for a total of 3 different genetic lineages?

Answer – Got corrected

If you want to look at subgroupings within your two clusters, I think the appropriate thing to do is to look at the results from your other K=X runs. If there really are 5 "sub-groups" in your data set, then you should see this in the K=5 run and you can present those results. If you don't see that in the K=5 run, I think that reflects that your "sub-groups" are just artefacts of the weird tiered STRUCTURE design you have come up with. I am almost certain that you will find that your 5 sub-groups are artefactual based on the results for Cluster 2: when you run STRUCTURE on just those individuals, you get two clusters where every individual has about half its genetic background from each cluster. This is the result you get when you force-feed a single panmictic population into STRUCTURE: STRUCTURE "analyzes" the population by just splitting it right down the middle like that, and the Evanno method chooses k=2 because the Evanno method is incapable of evaluating (the correct) k=1. So I don't think you actually have any population structuring within Cluster 2 (at least, not that STRUCTURE is able to detect). Finally, I would recommend that you give us the results of the Evanno analysis, especially if you are going to ignore it and explore K=3, K=4, K=5, etc. That's not necessarily a bad thing to do (the Evanno method is mostly a rule of thumb), but the results of the Evanno analysis will help us figure out how valuable this is. If the delta-ln-whatever values for K=2, K=3, K=4, etc are all very similar, then that will support your desire to look at a bunch of different K-values. But if the value for K=2 is by far the best, that will be more evidence that there isn't much structure in the population apart from the two clusters you have identified.

Answer - We repeat analyses with the Evanno method identified 2 as the optimal number of clusters (k=2), see, Figure 2.

Second bad STRUCTURE thing:

STRUCTURE is coming too late in this analysis. You are using STRUCTURE to identify a bunch of "clusters" that have lots of gene flow within them but are genetically distinct from other clusters. This is a great use of STRUCTURE. However, this should happen BEFORE you do the rest of population genetic analysis. I.e., if FLOR/TE/PAR/TI represent a single panmictic population, it doesn't make sense to calculate Na, Ap, Ho, FIS, etc, for each one of those sampling sites separately; instead you should combine the individuals into a single "Piaui" population and calculate your pop-gen stats using all the individuals together. This will have the additional benefit of making Tables 3 and 4 much smaller (since you'll only have 2-3 populations) and easier to read and interpret. (If you want to keep the original tables with the site-by-site stats in the SI, I'd be OK with that).

Answer - We accept the reviewer's guidance. Got corrected

Third bad STRUCTURE thing:

Speaking of identifying clusters: First, I recommend that you not use the term "clusters". This is confusing because you mean clusters-of-populations, but STRUCTURE is a clustering algorithm and the results that it produces are also "clusters" (i.e., all the red in Figure 2 is a "cluster"). You don't use "cluster" in this latter sense, but people who are familiar with STRUCTURE, etc, do, and so it makes things confusing.

I recommend that you change your terminology like this: BC, SD, GEB, etc, should be called "sites". Then you can divide those "sites" up into "populations" (what you are now calling "clusters") based on the results of your STRUCTURE analysis. I think that this terminology does a better job of capturing the genetic reality that we are discussing, so I am ging to use it for the rest of this comment.

You never talk about how you divide the sites into clusters, and you imply that this division was an output of STRUCTURE. This simply isn't true; STRUCTURE doesn't cluster individuals or sites into populations (or at least, the mode of STRUCTURE you ran doesn't do that). Instead, defining populations is a bit of an art in which STRUCTURE data is useful, but you are still making qualitative judgements that you need to describe and justify to the reader. For instance, here is how I would do it: "STRUCTURE identified two genetic lineages; based on these results, we divided our sampling sites into two populations: the West population included those sites in which the first of the two lineages predominated (the red bars in Figure 2), i.e., BC, SD, SAC, and VG. The East population included those 

sites in which the second of the two lineages predominated (the green bars in Figure 2), i.e., GEB, TI, TE, PAR, and FLOR".

Answer – Got corrected see lines 201-206.

The weird thing here is that GEB looks like a Piaui population that has very little red heritage, even though it is right between two populations that are dominated by red. Regardless of whether you want to put GEB in a population with the other Maranhao sites or with the Piaui sites, I think you need to try to explain what is going on here: is there some geographic feature like a river that might be separating GEB from SD and SAC?

MODERATE THINGS: Most of these should be changed or addressed.

50: I believe the authority for Ae. aegypti is (Linnaeus, 1762), no italics. Linnaeus put the species in Culex, hence the parentheses.

Answer – Got corrected, see line 45.

76-91: I think it would be worth describing how your study differs from the previous studies in this region, either here or in the discussion. Are you just replicating their studies? Do you have more samples, more markers?

Answer – Got corrected, see lines 73-79 and 251-255.

143, 163: It is awkward that you have two Ne variables. Can you rename them Ne(allele) and Ne(pop) or something like that? Answer – Got corrected, see lines 127; 142; Table 3 and Table 7.

176: I think you mean "per locus, per population", correct? You had more than 1 allele across all populations for all six of your microsats.

Answer – Got corrected, see line 150.

213-216: You mention some results here, but the data are never presented anywhere that I could find.

Answer – Got corrected, that part has been removed from the manuscript.

241-242: I don't think that the numbers in parentheses are correct: surely both 85%s should be in the first parentheses, and the second parentheses should be the ~14% numbers?

Answer – Got corrected

Table 2: This needs a caption describing what F, pb, etc mean.

Answer – Got corrected

Table 3: It would be nice to know the total number of alleles found across all 9 populations. Also, there is no explanation of what "r" means in the caption. Finally, the "Na" values you give at the bottom across all 6 microsats: you have calculated them as means, not medians (and perhaps done the same for Ne and Ar, though I didn't check).

Answer – Got corrected

Table 4: I recommend you make it easier to find the significant values in this table: maybe bold them? 

Answer – Got corrected

Table 6: There is a value in the BC/BC cell: what is that? Also, there are two values in the TI/TE and FLOR/TE cells above the diagonal--is this just a problem with the typesetting of the table, or are there actually two different values there?

Answer – Got corrected

Table 7: You might want to clarify in the caption whether these are the 6 Maranhao sites vs the 3 Piaui sites, or whether Maranhao means "Maranhao minus Timon", etc.

Answer – Got corrected

Figure 1: I think that the text in this figure should be in English (or in many cases it can probably be removed altogether). Why are there two insets of Brazil? I would delete the lower of the two; this would give you more space to make the key easier to read. I would encourage you to use different symbols for the two clusters you find; then it is easy for the reader to visualize how the TI site groups with the Piaui sites. Also, you refer to Piaui and Maranhao several times in the text, I think it would be valuable to put those names on the map here for the sake of your non-Brazilian readers (or your Brazilian readers who are bad at geography). Finally, I recommend that you explain in the caption that Mid-North is the shaded region; it's in the key, but not entirely obvious to me.

Answer – The Figure 1 should was redrawn.

Figure 2: You need to include what numbers 6-9 stand for.

Answer – Got corrected

MINOR THINGS: I think these would improve the ms, but if you don't want to do them, that's fine.

32: Maybe "four very important arboviruses"? There are a lot more than these that it also vectors, no?

Answer – Got corrected

34: I would write "control measures"

Answer – Got corrected

53: You can lose the abbreviations (ZIKV, etc) as you never use them later on in the paper (you do the correct thing in spelling out the diseases later on).

Answer – Got corrected

94: this "region is a sub-region of Northeastern Brazil" sounds strange. Maybe just delete the "sub-" part?

Answer – Got corrected

107: "4-7 days"?

Answer – Got corrected, see lines 96-100.

108: I'm not sure "pallets" is colloquial English, although without seeing your ovitraps I'm not sure what term to use--if they lay eggs onto paper, I might just call them "papers".

Answer - Palettes are small plywood boards with a suitable surface for female eggs to adhere to the ovitraps.

114: "microtubes"?

Answer – ? Yes, Got corrected

117: I would say "manufacturer's instructions" (like you do later in the paragraph).

Answer – Got corrected

140: You use "verify" in several places to mean "test for" or "analyze" or similar. I think you should restrict "verify" to mean "confirm something that we already believed to be true", which I don't think is what you mean here.

Answer – Got corrected

156: I think you mean "repetitions" or something like that, not "interactions".

Answer – Got corrected

Table 2: I would say "Repeat type"

Answer – Got corrected

Round 2

Reviewer 3 Report

I think the manuscript has been improved to the point that it can be accepted for publication.

Please note that there are some formatting issues with tables that should be checked at copyediting.

Author Response

I think the manuscript has been improved to the point that it can be accepted for publication.

Please note that there are some formatting issues with tables that should be checked at copyediting.

Answer - The formatting issues with tables that was checked.

Reviewer 4 Report

Looking good, I think the paper had been much improved by the changes. A couple of things I noticed:

You might want to note somewhere in Table 1 or a caption that TI (and GEB?) are in the "state" of Maranhao but in the "population" of Piaui.

Also, in the analyses presented in table 7 and the text in 170-173, you have GEB in the Maranhao population, but in the AMOVA analyses presented in table 5 you have GEB in the Piaui population.

I don't think I have a very strong opinion about which population you put GEB into, but I think that you should be consistent across the analyses and you might want to add a sentence or two to the text (around line 170, perhaps) explaining why you chose whichever approach you chose.

Author Response

Looking good, I think the paper had been much improved by the changes. A couple of things I noticed:

You might want to note somewhere in Table 1 or a caption that TI (and GEB?) are in the "state" of Maranhao but in the "population" of Piaui.

Answer - In the table 1 was note a caption that TI and GEB are in the "state" of Maranhao but in the "population" of Piaui.

Also, in the analyses presented in table 7 and the text in 170-173, you have GEB in the Maranhao population, but in the AMOVA analyses presented in table 5 you have GEB in the Piaui population.

Answer - This information corrected in the text, GEB belong in the "population" of Piauí, see lines 171-174.

I don't think I have a very strong opinion about which population you put GEB into, but I think that you should be consistent across the analyses and you might want to add a sentence or two to the text (around line 170, perhaps) explaining why you chose whichever approach you chose.

Answer - Was add a sentence to the text, see lines 174-176